# Infection of humanized mice with a novel phlebovirus presented pathogenic features of severe fever with thrombocytopenia syndrome

Shijie Xu[1], Na Jiang[1], Waqas Nawaz[1], Bingxin Liu[1], Fang Zhang[1], Ye Liu[1], Xilin Wu[1]*, Zhiwei Wu[2,3,4]*

1 Center for Public Health Research, Medical School, Nanjing University, Nanjing, China, 2 School of Life Sciences, Ningxia University, Yinchuan, P.R. China, 3 State Key Laboratory of Analytical Chemistry for Life Science, Nanjing University, Nanjing, China, 4 Jiangsu Key Laboratory of Molecular Medicine, Medical School, Nanjing University, Nanjing, China

* xilinwu@nju.edu.cn(XW); wzhw@nju.edu.cn(ZW)

## Abstract

Severe fever with thrombocytopenia syndrome virus (SFTSV) is a tick-borne emerging phlebovirus with high mortality rates of 6.0 to 30%. SFTSV infection is characterized by high fever, thrombocytopenia, leukopenia, hemorrhage and multiple organ failures. Currently, specific therapies and vaccines remain elusive. Suitable small animal models are urgently needed to elucidate the pathogenesis and evaluate the potential drug and vaccine for SFTSV infection. Previous models presented only mild or no pathogenesis of SFTS, limiting their applications in SFTSV infection. Therefore, it is an urgent need to develop a small animal model for the investigation of SFTSV pathogenesis and evaluation of therapeutics. In the current report, we developed a SFTSV infection model based on the HuPBL-NCG mice that recapitulates many pathological characteristics of SFTSV infection in humans. Virus-induced histopathological changes were identified in spleen, lung, kidney, and liver. SFTSV was colocalized with macrophages in the spleen and liver, suggesting that the macrophages in the spleen and liver could be the principle target cells of SFTSV. In addition, histological analysis showed that the vascular endothelium integrity was severely disrupted upon viral infection along with depletion of platelets. *In vitro* cellular assays further revealed that SFTSV infection increased the vascular permeability of endothelial cells by promoting tyrosine phosphorylation and internalization of the adhesion molecule vascular endothelial (VE)–cadherin, a critical component of endothelial integrity. In addition, we found that both virus infection and pathogen-induced exuberant cytokine release dramatically contributed to the vascular endothelial injury. We elucidated the pathogenic mechanisms of hemorrhage syndrome and developed a humanized mouse model for SFTSV infection, which should be helpful for anti-SFTSV therapy and pathogenesis study.

**Data Availability Statement:** All relevant data are within the manuscript and its Supporting Information files.

**Funding:** This work was supported by National Science Foundation of China (NSFC) (No. 81803414 to XLW, 31970149 to ZWW), The Major Research and Development Project (2018ZX10301406 to ZWW), Jiangsu Province Natural Science Foundation for Young Scholar (grant #BK20170653 to XLW), Nanjing University-Ningxia University Collaborative Project (grant #2017BN04 to ZWW), Jiangsu Province "Innovative and Entrepreneurial Talent" project, Six Talent Peaks Project of Jiangsu Province and Fundamental Research Funds for the Central Universities (021414380341 and 021414380432 to XLW). The funders had no role in study design, data collection and analysis, decision to publish, or preparation of the manuscript.

**Competing interests:** The authors have declared that no competing interests exist.

## Author summary

SFTSV is a novel bunyavirus that was identified in 2010 and endemic in China, Korea, Japan and Vietnam with expanding spatial incidents. SFTS is characterized by high case-fatality rates and currently has no effective therapeutics or vaccines. In previous study, models presented only mild or no pathogenesis of SFTS, limiting their applications in SFTSV infection. In the current study, we developed a humanized NCG mouse model for the study of SFTSV infection and elucidated the pathogenic mechanisms of hemorrhage syndrome with respect to apoptosis, membrane protein endocytosis and cytokine stimulation. The HuPBL-NCG model presented multiple organ pathologies that resemble those of human infection, which will be helpful for anti-SFTSV therapy and pathogenesis study.

## Introduction

Severe fever with thrombocytopenia syndrome (SFTS) is a recently identified emerging infectious disease caused by a novel phlebovirus in the family Phenuivirida of the order Bunyavirales [1]. SFTS virus (SFTSV) was reported and isolated in China in 2010 [2,3], followed by South Korea, Japan and recently Vietnam, with increasing number of infection cases [4–8]. Other tick-borne *Phleboviruses* include Lone Star virus and Heartland virus isolated in the United States [9,10], indicating that phlebovirus infection may have a broad spatial presence. The clinical manifestations of SFTS are characterized by high fever, gastrointestinal symptoms, thrombocytopenia and/or leukocytopenia, hemorrhage and multi-organ failures. Patients die from multiple organ dysfunction and disseminated intravascular coagulation accompanied by hemorrhage [11,12] with a fatality rate ranging from 6% to 30%. However, the mechanism of hemorrhage caused by SFTSV remains unclear. SFTS has been listed as one of the emerging infectious diseases requiring priority research and intervention by the World Health Organization (WHO).

Although there are a number of small animal models reported, most of them supported SFTSV infection with only limited or no pathogenic resemblance to human infection. One of the major symptoms caused by SFTSV infection, hemorrhage, was not observed [13–18]. Both alpha/beta interferon receptor knockout (IFNAR$^{-/-}$) mice and mitomycin-treated mice are highly susceptible to SFTSV infection with 100% animals succumbing to the infection within 3 to 4 days. The short survival time limited the use of this model for studying the pathogenesis and for evaluating therapeutics. Furthermore, SFTSV infection in immunocompetent animals, as hamsters and C57/BL6 mice, resulted in reduction of white blood cells and platelets, but did not cause severe symptoms or death, indicating that immunocompetent animals did not accurately mimics the pathogenesis and clinical manifestations of SFTSV infection.

Aged-ferrets effectively mirror characteristics of human infection, including high fever, severe thrombocytopenia, viremia, and body weight loss, but young adult ferrets do not exhibit clinical signs of SFTS. Besides, lethal SFTSV infection of aged-ferrets quickly resulted in death in 6 to 8 days, which only provide a brief window period to study the pathogenesis, such as leukocytopenia, haemorrhage and multiple organ failure.

In this study, we reported the development of a humanized mouse model by engrafting NCG mice with human PBMCs. A number of key clinical features of SFTSV infection could be mimicked in this mouse model. This model not only illuminated the main symptoms of SFTS infection, but also could be a promising candidate for systematic investigation of the viral pathogenesis and the efficacy of antiviral drugs.

## Results

### SFTSV infection of HuPBL-NCG mice

HuPBL-NCG mice were constructed by intraperitoneal injection of human PBMC into NCG mice and examined the susceptibility of the animals to SFTSV infection. Multi-lineage human hematopoiesis was demonstrated by FACS analysis of lineage-specific markers. $CD3^+CD4^+$ T cells, $CD3^+CD8^+$ T cells and $CD45^+CD14^+$ monocytes were present in reconstituted mice (Figs 3A and S1A), suggesting that HuPBL-NCG mice harbored the major cellular targets for SFTSV infection. NCG, along with other mouse strains (BALB/c, C57BL/6, AG6) were infected with the SFTSV E-JS-2013-24 strain at $10^5$ $TCID_{50}$ per mouse and analyzed for pathogenic manifestations. SFTSV caused no abnormal clinical signs in NCG mice (S1C Fig) or in immunocompetent BALB/c and C57BL/6 mice [16], whereas SFTSV resulted in 100% lethality in 4 days post infection in immunocompromised interferon α/β and γ receptors deficient AG6 mice (S1B Fig). Next, groups of 9-week-old humanized NCG mice ($n$ = 5) were inoculated intraperitoneally (i.p.) with three different doses (low = $10^3$ $TCID_{50}$, medium = $10^4$ $TCID_{50}$, high = $10^5$ $TCID_{50}$) of SFTSV and blood was collected through the orbital sinus on days 1, 3, 5, 7, 14, and 35 post infection for determination of viral RNA copies (Fig 1A). The results showed that blood viral load increased rapidly in the first week for all three viral inocula and increased gradually from the $2^{nd}$ week up to the $5^{th}$ week when the animals were sacrificed (Fig 1B). Our results showed that HuPBL-NCG mice supported productive infection leading to sustained viremia, which resembles the clinical infection.

SFTSV was detected in multiple organs in our humanized mice model, indicating a systemic infection [19,20]. The viral distribution and replication in heart, liver, spleen, lungs, kidneys, stomach, intestine, brain and muscle tissues were investigated in infected mice. SFTSV was detected in all organs or tissues of the mice infected with low, medium and high dose of virus (Fig 1C). The spleen and liver showed the highest viral loads per milligram tissue in high and medium groups. Furthermore, we also detected virus in brains of all three groups, suggesting central nervous system dissemination (Fig 1C).

All SFTSV-infected animals exhibited 10–30% weight loss by 28 days post infection, as compared to the negative group (Fig 1D). The mice infected with higher titer virus showed increased mortality and early death over the course of the experiment (Fig 1E) while the body temperature during SFTSV infection of HuPBL-NCG mice did not present an obvious variation (S1D Fig). In addition, we observed that SFTSV infection of HuPBL-NCG mice caused reduction of platelets on day 1 post infection that gradually decreased by day 14 post infection (Fig 1F), along with a gradual decrease in white blood cell counts continued until death (Fig 1G). Thus, these results indicate that HuPBL-NCG mouse model can support SFTSV replication and recapitulate the tissue tropism as in human infection.

### Histopathological lesions in SFTSV-infected humanized mice

We analyzed the pathological lesions in various organs and tissues by H&E staining. Tissues were collected from sacrificed animals and pathological changes were identified in the spleen, lungs, kidneys and livers. The number of lymphocytes in the red pulp was visually decreased in spleen, a secondary hematopoietic organ in mice, of SFTSV infected mice (Fig 2A). This lymphocyte depletion in the red pulp may be consistent with systemic reduction of white blood cells. Besides, we observed a remarkable increase of megakaryocytes infiltration in the spleen based on their cellular morphology (Figs 2B and S2A). This result supports extramedullary hematopoiesis, which has been reported to occur in conjunction with decreased lymphocyte cellularity of the red pulp [21]. Since megakaryocytes are progenitor cells for platelets, it is

possible that the rapid increase of megakaryocytes in the hematopoietic organs of the spleen functioned as compensation for the depletion of circulating platelets. Therefore, once mega-karyocytes have proliferated, they would persist in organs for a long period [22]. Peribronchio-lar inflammation was evident with alterations in bronchiolar cell structure found throughout the lungs. Significant interstitial infiltration was observed with peri-vascular cuffing and extensive alveolar thickening (Fig 2C). A number of small alveoli fused, which would greatly reduce the efficiency of oxygen exchange. The pathological changes were also noted in liver. The pathological lesions in liver consisted of ballooning degeneration of hepatocytes and scattered necrosis, the latter indicated by multifocal pyknosis, karyorrhexis, and karyolysis (Fig 2D). The kidneys showed congestion in capsular space, mesangial thickening, and glomerular hypercellularity (Fig 2E). Importantly, these pathologic lesions were consistent with the findings of multiple organ failure in severe SFTS [23], suggesting that the humanized *in vivo* model of SFTSV infection recapitulates pathological consequence that was seen in SFTS patients.

## HuPBL-NCG mice increased SFTSV susceptibility and promoted virus infection of mice tissues

The percentage of monocytes infected by SFTSV was universally increased after virus inoculation and the intensity of viral antigen-positive monocytes increased markedly in the higher $TCID_{50}$ groups. FCM showed that monocytes were the major cell type infected by SFTSV among CD45 positive cells (Fig 3A and 3B), consistent with human infections [24].

Immune competent mice have been shown to be largely refractory to severe infection and disease following SFTSV challenge [16]. The wide presence of tissue pathology in HuPBL-NCG mice suggests that murine cells may be infected. There is a possibility that the infected human monocytes may transmit the virus to murine cells more efficiently through cell-mediated mode. We thus conducted co-culture assay with THP-1, a human monocyte cell line, and RAW264.7, a murine macrophage-like cell. We performed analysis by flow cytometry on infected THP-1 double-stained with CD45-APC and Gn-FITC, and showed that THP-1 cells were almost always 100% infected by SFTSV in our experiments (S3 Fig). It was also demonstrated that almost all RAW264.7 cells infected with SFTSV in THP-1 and RAW264.7 co-culture, in contrast to 68.62% infection when RAW264.7 was infected alone. Fig 3C immuno-fluorescence also confirmed this result by staining SFTSV Gn protein in co-culture and single culture. QPCR results showed that the Gn expression in RAW264.7 cells increased significantly, as compared with that of RAW264.7 infected alone or in the presence of exogenous proinflammatory cytokines (TNF-α and IL-1β) (Fig 3D). These results revealed that THP-1 facilitated the viral infection of RAW264.7 cells *in vitro* via direct cell-cell contact. In addition, in part of the SFTSV infected murine tissues, Gn positive human monocytes were found by fluorescence staining assay (Fig 3E). These results suggested that permissible human cells in HuPBL-NCG mice probably transmitted the virus to murine monocytes through cell-cell contact *in vivo*.

## Virus localization and proinflammatory cytokine determination in infected animals

To determine the specific SFTSV-infected cells in murine tissues, we examined colocalization of SFTSV antigens with various tissue cells, using confocal microscopy on splenic tissues of SFTSV-infected HuPBL-NCG mice. Viral RNA was enriched in the spleen and liver with marked pathological changes in these organs during the late stage of infection, showing that the spleen and liver likely had the highest SFTSV load among organs of the infected animals

(Fig 1C). Besides, *in vivo* imaging assay was performed by NightOwl LB 983 at the indicated time points after fluorescence labled Gn specific antibody intravenous injection in mice, which confirmed that SFTSV was accumulated in the liver, spleen, lung, stomach and intestine (S2B Fig). Deparaffinized tissues in thin sections were double stained with antibodies to SFTSV Gn and macrophages (anti-F4/80). Confocal microscopy showed that Gn colocalized with F4/80, suggesting that murine macrophages of the spleen and liver were infected (Fig 4A and 4B). In addition, we observed Gn staining in brain and kidney, indicating that SFTSV also infected these organs (Fig 4C and 4D). Based on these observations, we postulated that the macrophage-agminated spleen and liver were the primary target organs of SFTSV.

In the acute phase of SFTSV infection, disease severity correlates with a cytokine storm. In clinical investigations, including our patient cohort study, several cytokines were previously identified to be key biomarkers of disease severity [25,26]. Therefore, we examined cytokine production in SFTSV-infected HuPBL-NCG mice throughout the experimental infection course (Fig 4E) and found that, in infected mice, the production of IL-6 was significantly elevated during the early stage of infection and both TNF-α and IL-1β were significantly elevated during the late stage of infection, highly consistent with observations in human infection [12]. The augmentation of inflammatory cytokines during the experimental infection course indicated that SFTSV infection induced systemic host antiviral immune responses in HuPBL-NCG mice.

## SFTSV infection disrupted the endothelial cell barrier

Haemorrhage is a hallmark of SFTS pathology [23] and the depletion of platelets is considered a critical factor [16]. In addition to the virus-induced thrombocytopenia in infected HuPBL-NCG mice, we observed disorganization of vascular endothelium in sacrificed mouse lung sections by immunostaining with anti–platelet cell adhesion molecule 1 (PECAM-1; CD31) antibody and H&E staining (Figs 5A and S4). The continuity and integrity of endothelial cells markedly decreased as campared with uninfected control mice. Concomitantly, we found a higher incidence of red blood cell leakage around the disintegrated pulmonary vessel as shown in Figs 2 and 5A, suggesting that SFTSV infection impairs endothelial function and results in vascular barrier disruption. Transwell assay showed the increased permeability of HUVECs monolayer with increasing virus inoculum (Fig 5C). No evident difference was found between various doses of virus inoculum in scratch migration assay (Fig 5D), while a marked migration delay was observed after SFTSV infection, and permeability change was also apparent with the viral challenge. Haemorrhagic symptom was considered as a higher risk of death among clinical patients [23]. Increased efflux of fluid and macromolecules from the intravascular space leads to profound tissue edema, hypotension and shock [27,28]. Consequently, we estimated the effect of SFTSV on the permeability of blood vessels. By using the Miles assay, the extravasation of Evans Blue dye from vessels in the lungs of a mouse can be quantified. We found that virus infection had a promotive effect on microvascular permeability in a dose-dependent manner, which corresponded to the permeability change in transwell assay and quantitative analysis revealed that about 3.2-fold increased leakage occurred in lungs infected with SFTSV as compared with lungs injected with saline only (Fig 5B).

We also investigated the effect of SFTSV on tubule formation of HUVECs on Matrigel. HUVECs cultured on Matrigel go through morphological changes with formation of a tubule, of which in possession a lumen surrounded by endothelial cells adhering to one another. We indicated that SFTSV inhibited tubule formation of HUVECs, while stimulated by VEGF, as quantified with NIH Image software by pixels (Fig 5E).

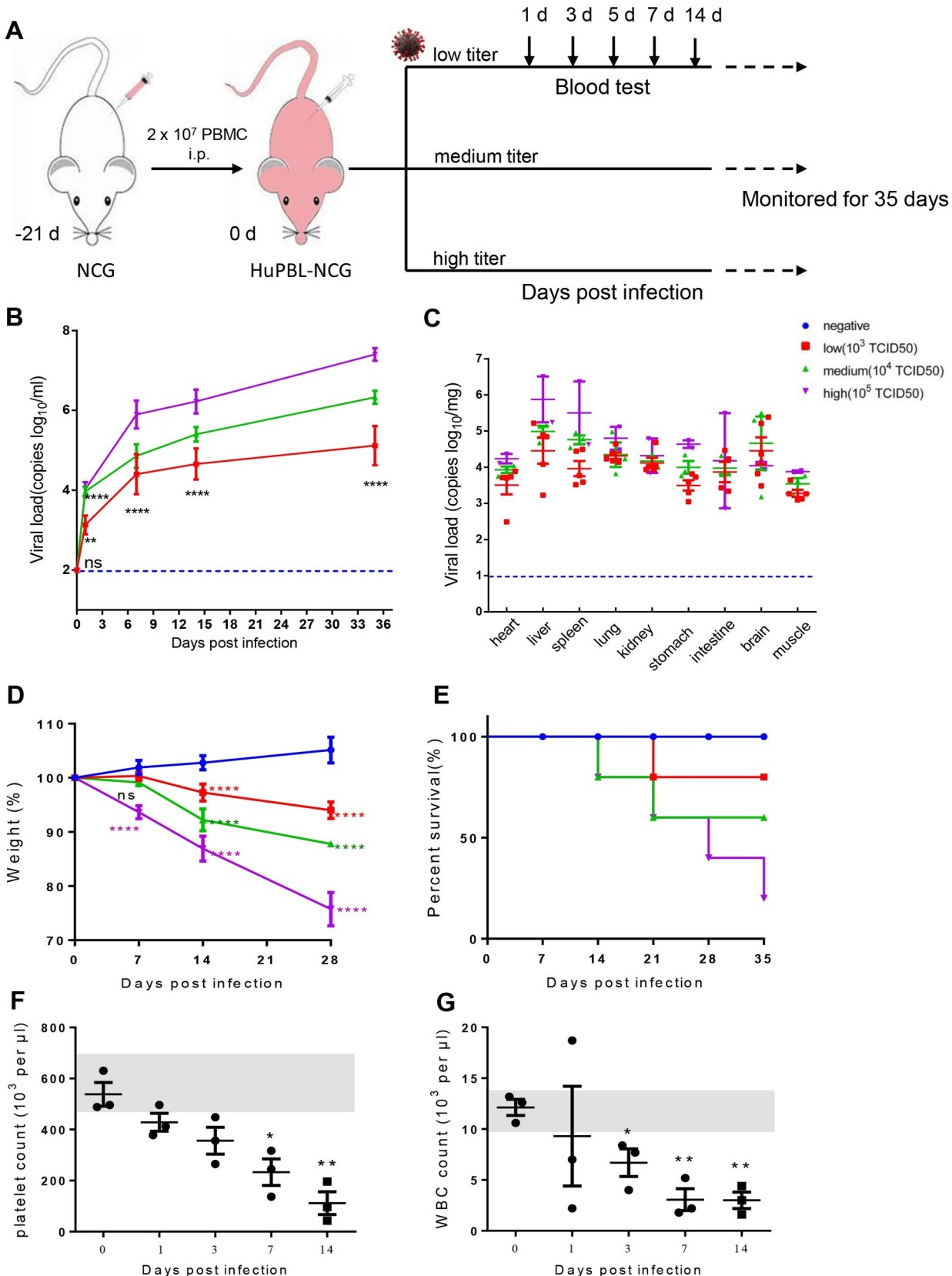

**Fig 1. Development of a humanized animal model for SFTSV infection.** Clinical manifestations and haematological analysis of SFTSV-infected HuPBL-NCG mice. (A) Humanized experimental schedule. (B and C) Distribution of the number of viral RNA copies in blood and tissues of SFTSV-infected mice. Blood samples ($n = 5$ per group) were collected at 1, 3, 5, 7, 14, or 35 dpi. and tissues ($n = 3$ per group) were collected on sacrifice day. Relative weight and survival were assessed and are shown as standard error of the mean. Data (mean ± SEM) are presented in D and E. The blood from infected mice was collected at 0, 1, 3, 7 or 14 days and haematological examination was performed using a hematology analyzer. Platelet counts (F) and WBC counts (G) from high dose virus infected HuPBL-NCG mice are shown. Normal ranges of platelet and WBC counts in uninfected mice are $480–700 \times 10^3$ per μl and $9.5–14.2 \times 10^3$ per μl, respectively. The gray area indicate the normal values of platelet or WBC counts. The asterisks indicate significance between non-infected and infected mice dpi as determined by a two-tailed unpaired $t$-test. Data are shown as mean±SEM of three independent experiments.($^*$p <0.05, $^{**}$p <0.01, $^{****}$p <0.001).

Taken together, the above results indicate that SFTSV not only disrupted endothelial junctional integrity but also inhibited the proliferation, chemotaxis, and tubule formation of endothelial cells, contributing to the haemorrhagic outcome of the infection.

## Apoptosis and VE-cadherin internalization were enhanced in SFTSV-infected endothelial cells

Virus-induced vascular leakage and hyperpermeability could be caused by programmed cell death [29]. To measure cell apoptosis, HUVECs were stimulated with various doses of virus (MOI 1 and 10) from 24 to 120 h. Virus-induced apoptosis was observed at 48 h in 10 MOI group post infection (Figs S5A and 6A). Direct apoptosis measurement using TUNEL/Caspase 3 immunofluorescence staining was also conducted in lung section. The results showed that endothelial apoptosis could be observed in virus infected mouse model(S5B Fig) However, there was no significant virus-induced apoptosis in low-dose SFTSV infection during 5 days of infection (S5C Fig), suggesting a dose dependent apoptotic effect in SFTSV infected HUVECs.

To evaluate the immunopathology of SFTSV on vascular endothelial cells and monocytes, the levels of IL-1β, IL-6 and TNF-α, inducers of proinflammation, in the culture supernatants, were analyzed. Pronounced elevation of IL-1β and IL-6 was induced by SFTSV infection (Fig 6C and 6D), consistent with observations in SFTS patients [26]. Fig 6F indicated that either SFTSV alone or SFTSV-inducible inflammatory cytokines could induce endothelial hyperpermeability as shown *in vitro*. Using the SFTSV permeability assay we further demonstrated that an TNF-α inhibitor (Lenalidomide), IL-1β inhibitor (AS101) and MOSLOFLAVONE, an inhibitor of both cytokines, suppressed SFTSV-induced permeabilizing responses at physiologic concentrations (S5E Fig).

Vascular endothelial cadherin (VE-cadherin) is an endothelial cell-specific adherens junction protein and the primary determinant of paracellular permeability within the vascular endothelium [30–32]. Tyrosine 685(Tyr$^{685}$) phosphorylation of VE-cadherin is assumed to affect endothelial junction integrity [33]. We found that SFTSV induced phosphorylation of VE-cadherin at Tyr$^{685}$ from 24h post infection (Fig 6B). The internalization of VE-cadherin was reported to regulate barrier functions of endothelial cell adherens junctions [34]. In the absence of SFTSV, VE-cadherin is present on the surface of cytokins- and mock-treated endothelial cells (Fig 6E, Acid-), since acid washing completely removed VE-cadherin from the surface of cells (Fig 6E, Acid+). However, following SFTSV infection and subsequent acid washing, there is a dramatic increase in the level of intracellular VE-cadherin (Acid+) within endothelial cells, especially in cells treated with the combination of inflammatory cytokines. (Fig 6E, SFTSV+IL-1β+TNF-α). In contrast, SFTSV addition to mock- or cytokine-treated cells markedly promoted the VE-cadherin internalization in endothelial cells, especially enhanced the response to inflammatory cytokines, demonstrating that not only SFTSV infection, but also cytokines have disruptive effects on the endothelial integrity resulting in capillary leakage, and internal haemorrhage. The result was also confirmed in S5D Fig, which showed

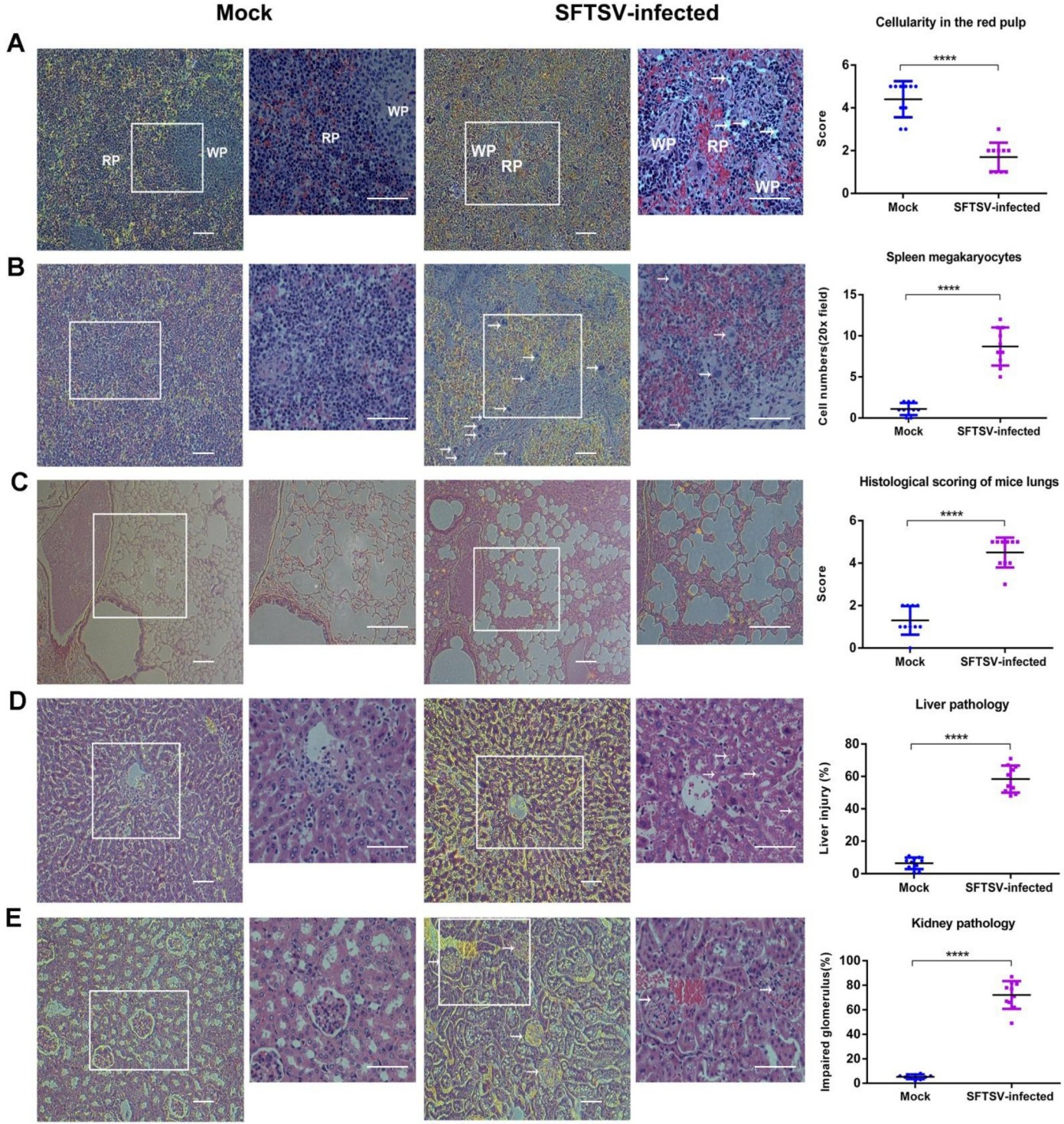

**Fig 2. Histopathological lesions in SFTSV-Infected humanized mice.** Dynamic profile of pathological changes in mice infected with SFTSV. Representative H&E-stained tissue sections from SFTSV-infected mice and mock mice. Areas of interest (AOI) are enlarged, and quantitative graphs are presented at the right. (A) Decreased cellularity and obvious lymphocyte depletion in the red pulp (RP) of the spleen at sacrificed day. (B) The number of megakaryocytes increased significantly in the spleen. (C) According to the inflammation of pulmonary interstitium, the thickening of alveolar septum and the structural integrity of pulmonary vesicles, the scores are divided into five grades. In the C images and AOI, structure of the alveoli was obviously destroyed, and the alveoli were filled with inflammatory substances. There was no obvious alveolar septum. (D) In liver, arrowheads show hepatocyte necrosis with shrinking nucleic or hepatocyte degradation with balloon-like empty cytoplasm. (E) In the kidney, arrowheads show impaired renal capsules. Bleeding was observed and the space between glomerulus and surrounding tissue was blurred. In figures, the images and AOI are 100× and 200×, respectively. Bar = 50 μm. In quantitative graphs, red dots and blue dots indicate SFTSV-infected mice and mock mice, respectively. Data are shown as mean±SEM of three independent experiments. (****p <0.001).

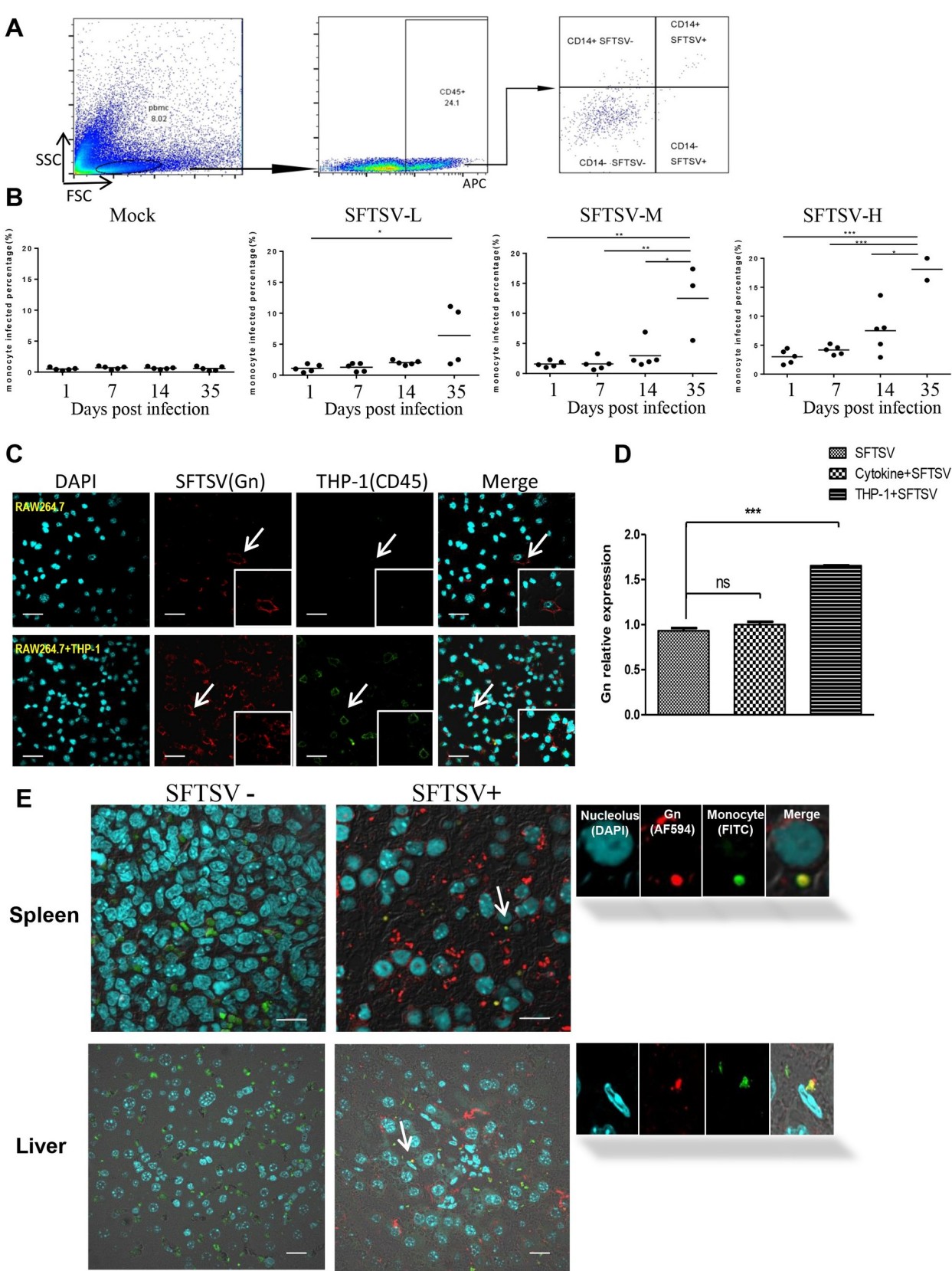

**Fig 3. HuPBL-NCG mice increase SFTSV susceptibility in host tissues.** (A) Flow cytometry analysis of the percentage of CD14$^+$ human monocytes among SFTSV infected cells. (B) At different time point, the percentage of virus-infected monocytes among infected with different virus titer in HuPBL-NCG mice. (C) THP-1 significantly improve the virus infection ability among mouse RAW264.7 cells *in vitro*. Bar = 50 μm. (D) Relative expression levels of viral mRNA in RAW264.7 cells in different co-cultured assays. (E) SFTSV infected cells in HuPBL-NCG mice detected by IFC on sacrifice day. Representative tissue sections of spleen and liver were analyzed by antibodies specific to SFTSV Gn in red and to human CD14 in green among three groups of mice. Nucleoli was stained by DAPI in cyan. Bar = 20 μm. The enlarged images of individual cells were pointed by corresponding arrows. The images are 600 x. Data are shown as mean±SEM of three independent experiments. (*p <0.05, **p <0.01, ***p <0.005).

that three anti-inflammatory inhibitors significantly increased the amount of VE-cadherin protein on the cell membrane.

## Antiviral drugs stabilized the vasculature and reduced mortality in SFTSV infected mice

To evaluate the potential use of the animal model for evaluating antiviral drugs, we selected two candidate molecules, anisomysin and vitexicarpin, from an *in vitro* screen of 40 natural drugs for anti-SFTSV activity on Vero-E6 cells. IC$_{50}$ values for anisomysin and vitexicarpin were 6.338 and 1000 nM, respectively (S6A–S6C Fig), with minimal cytotoxicity (S6D and S6E Fig). We measured the permeability changes of endothelial cells on HUVECs were treated with drugs at different concentration, the results showed that the permeability of endothelial monolayers significantly improved with the presence of these two drugs at 48 hours post infection (S6F and S6G Fig).

To evaluate the *in vivo* activity of these drugs, 7-week-old female HuPBL-NCG mice were challenged intraperitoneally with SFTSV, and at 1 hpi, 1dpi, 4dpi and 7dpi, intraperitoneally treated with anisomysin, vitexicarpin or mock (saline) for seven days. The viral load in the blood was significantly inhibited in the drug-treated but not in the mock-treated mice (Fig 7A). Both of the drugs markedly reduced vascular permeability in the lungs (Fig 7B) and decreased the plasma level of multi-human inflammatory factors (S7 Fig). Consistently, there was no significant change in the body weight of mice in the drug treatment groups (Fig 7C). In addition, the mice treated with drugs showed a lower level of alanine aminotransferase (ALT) and aspartate aminotransferase (AST), and creatine kinase (CK) descended to the normal physiological reference range [35] at the 8 dpi in comparison with saline-treated mice (Fig 7D–7G). More importantly, the pathology of lungs and kidney in drug-treated mice was significantly improved as compared with that in the saline-treated mice (Fig 7H).

## Discussion

SFTSV is a novel pathogenic phlebovirus in the Phenuivirid family [1]. Despite the increasing human infections and expanding spatial distribution [36], the pathogenesis of SFTSV remains poorly understood due to the lack of a proper experimental animal model. Previously, immunocompromised animals such as mitomycin-treated mice, interferon receptor-deficient (IFNAR$^{-/-}$) mice, newborn mice and aged ferrets have been used to develop models. However, these models presented certain caveats in that they either resist virus infection [16,18], are susceptible to lethal infection with very short clinical course [13,18] or exhibit minimal pathology [18]. In addtion, among immunocompetent laboratory animals, rhesus macaques and C57/BL6 mice did not show severe clinical symptoms or death after SFTSV infection [17]. In this study, we developed humanized NCG mouse model for SFTSV that recapitulates the pathogenesis of human infection. In SFTSV-infected humanized mice, the spleen and livers were the primary target organs due to the presence of large number of macrophages that are targeted by SFTSV [16]. In addition to these organs, SFTS virions were also found in brain and intestine,

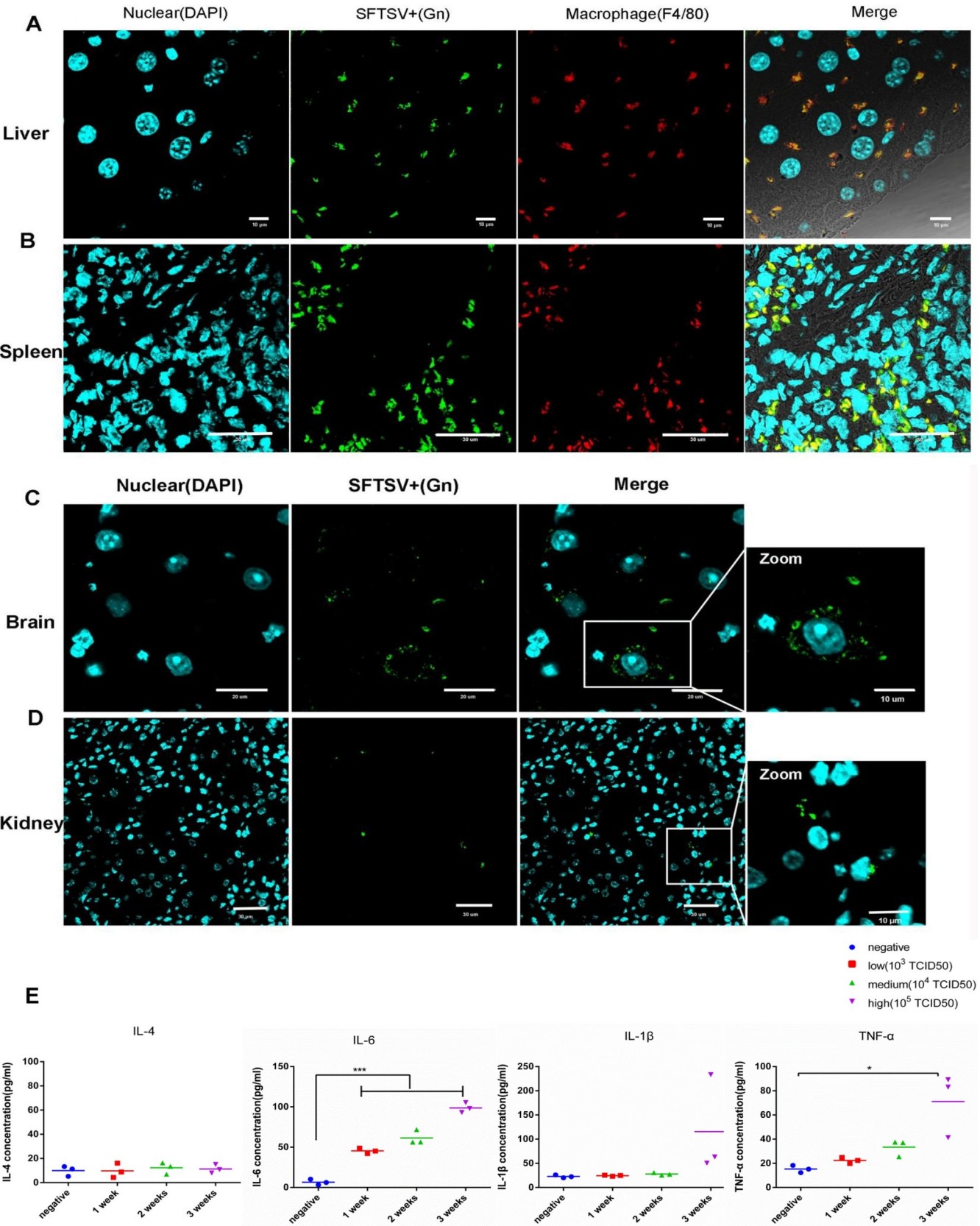

**Fig 4. Virus localization and proinflammatory cytokine determination in infected animals.** (A) SFTSV and macrophages were identified by immunofluorescence in the liver of SFTSV-infected mice. Experiments were performed with three independent trials and similar results were reproduced. Scale bars, 10 μm. (B). Confocal microscopy to examine colocalization of SFTSV (in green), and macrophages (in red) in the SFTSV-infected spleen. Scale bars, 30 μm. (C and D). The presence of the SFTSV Gn antigen was confirmed in the brain and kidney. Scale bars, 20 μm and 30 μm, respectively. (E). Kinetics of proinflammatory cytokines in SFTS virus–infected mice. Plasma levels of cytokines were quantified by ELISA assays between the infected group ($10^5$ TCID$_{50}$) and the control group (negative) after an indicated time. IL-1β, interleukin 1β; IL-4, interleukin 4; IL-6, interleukin 6; TNF-α, tumor necrosis factor α. Data are shown as mean±SEM of three independent experiments. (*p <0.05, ***p <0.005).

which may associate with gastrointestinal and neurological symptoms in severe clinical cases [23]. Elevated serum levels of AST, ALT, and CK indicated dysfunction in various organs, and pathological changes were observed in spleen, lung, liver and kidney during the late phase of SFTSV infection that were consistent with multiple organ failure in patients. Besides, in humanized mouse model, SFTSV infection caused thrombocytopenia (Fig 1F), leukocytopenia (Fig 1G), viremia (Fig 1B) and increased levels of inflammatory cytokines in blood, which are observed in severe cases of SFTS in humans. Notably, diarrhoea, neurological symptoms, dyspnoea, and haemorrhagic signs are important predictors of fatal outcome in independent studies [37,38].

Inoculation of SFTSV to NCG mice did not produce significant virus replication (S1C Fig), in contrast to the infection of HuPBL-NCG mice where robust viral replication and multiple clinical pathologies were observed. We conducted stringent quality control for the infection study after engraftment. Human PBMC was isolated from the blood of healthy donors provided by Jiangsu Provincial Red Cross. According to our previous reported method [39] in the construction of HuPBL mice model with following modifications: $2\times10^7$ PBMC from a single donor was injected intraperitoneally into each NCG mouse. Three weeks later, blood was taken from the mice to evaluate the percentage of human CD45. The mice with over 5% human CD45 in PBMC was considered as successfully humanized, which were then randomly divided into uninfected control group and infected group. The percentage of successfully humanized rate is basically over 90% and unsuccessfully humanized mice were discarded. Though the PBMCs isolated from different donors exhibited individual differences, HuPBL-NCG mice receiving human PBMCs from multiple donors were randomly divided into different groups to compensate for the inter-donor variation. Our results showed that the differences in SFTSV replication were not significant in the HuPBL-NCG mice receiving PBMCs from different donors.

The human PBMCs engrafted into NCG mice possibly provided an early replicating target for the virus and the infected human monocytes transmitted the virus to murine monocytes through cell-cell mediated mechanism, which is known to be much higher efficiency in viral infection [16,17]. In addition, infected human cells were infiltrating into the murine tissues, which may also contribute to the pathologies of various organs.

Haemorrhage is the hallmark of SFTS and 35% patients with a SFTSV infection had haemorrhagic sympotom, which is higher than that reported for other viral haemorrhagic fevers, such as Ebola virus disease or dengue [23,40,41]. Our study showed that SFTSV induced disorganization of vascular endothelium in the lungs of infected mice and red blood cell leakage around the disintegrated pulmonary vessels, spleen and glomerulus, suggesting that the HuPBL-NCG model can reproduce some key pathologic manifestations of SFTSV infection of humans. We believed that the virus-induced disruption of vascular endothelial integrity is a critical contributor to the hemorrhage while coagulation disorders due to platelet depletion alone should not be sufficient to lead to hemorrhagic outcome. In further support of this mechanism, we found that SFTSV infection significantly delayed the endothelial cell migration and tubule formation, thus affecting angiogenesis. Most importantly, high-dose SFTSV

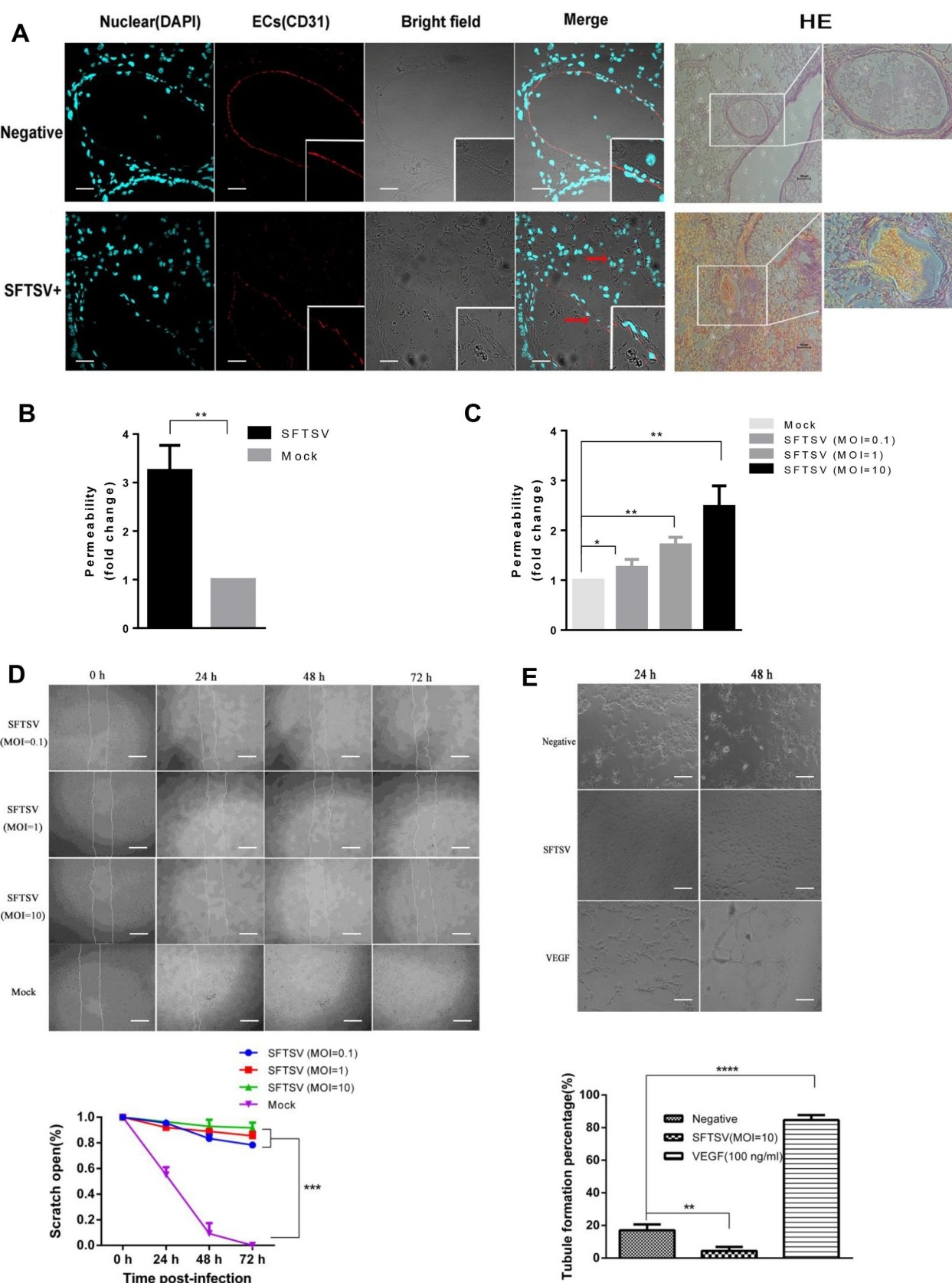

**Fig 5. SFTSV infection disrupts the endothelial cell barrier.** (A) Immunofluorescence staining and H&E staining lung tissues of mice infected with SFTSV on sacrifice day. The continuity and integrity of endothelial cells markedly decreased compared with negative control mice along with a higher incidence of red blood cell leakage around the disintegrated pulmonary vessel as shown by the red arrow. The fluorescent images and H&E images are 400× and 200×, Bar = 50 μm and 100 μm respectively. (B) The extravasation of Evans Blue dye from vessels in the lungs of a mouse can be quantified by Miles assay *in vivo*. Virus infection has a promotive effect on microvascular permeability with 3.2-fold increased leakage occurred in lung tissues compared with lung injected with saline only. (C) *In vitro* permeability was measured in HUVEC monolayer cells stimulated with different titers virus. (D) Scratch wound assay infected with SFTSV MOI = 0, 0.1, 1, or 10, respectively, was showed with time-points indicated. Original magnification was 40 x and quantifications of wound closure were shown in the graph below, bar = 125 μm. (E) Tubule formation of HUVECs were analyzed with the presence of SFTSV or VEGF. The results were quantified by counting the tube formation branch point as shown in the beneath graph, scale bars = 50 μm. Data are shown as mean±SEM of three independent experiments. (*p <0.05, **p <0.01, ***p <0.005, ****p <0.001).

infection directly induced apoptosis leading to the programmed cell death of HUVECs. In addition, the marked and abrupt release of multiple cytokines activated by the SFTSV further exacerbated the hemorrhage by destabilizing endothelial cell-cell interactions and crippled vascular barrier function through the internalization of VE-cadherin (Fig 6). In many infections, cellular factors generated from the cytokine-induced secondary inflammatory injury can be more toxic than the invading microbes themselves [42]. Our data indicate that SFTSV induced phosphorylation of VE-cadherin at Tyr[658] and SFTSV infection enhanced internalization of VE-cadherin in response to inflammatory cytokines, providing a mechanistic explaination to the vascular disruption. In particular, this study has revealed the machanism of hemorrhage that both SFTSV and SFTSV-induced cytokine storm are likely involved in inducing endothelial cells apoptosis and endocytosis of VE-cadherin, with the increased endothelial hyperpermeability (Fig 8).

We next evaluated two antiviral drugs against SFTSV *in vivo* using HuPBL-NCG mouse model. Both drugs inhibited viral replication (Fig 7A) and vastly reduced endothelial hyperpermeability in the blood vassels (Fig 7B), which mitigated the virus-induced pathology in lungs and kidney of the infected mice (Fig 7H). Most of the biochemical parameters descended to the normal physiological reference range in contrast to those in control animals (Fig 7D–7G). Consistent with the current observation, evaluation of an SFTSV Gn-specific single domain antibody showed inhibition of viral replication *in vivo* and mitigation of the virus-induced pathogenesis [39]. We have provided the first *in vivo* evidence that HuPBL-NCG could be used for pathogenic study of SFTSV infection and for evaluating therapeutics against SFTSV.

In conclusion, we have developed a humanized NCG mouse model for the study of SFTSV infection and pathogenesis. The HuPBL-NCG model presented multiple organ pathologies that resemble those of human infection. In addition, SFSTV replicated continuously in the animals up to 5 weeks and even in the high dose group ($10^5$ TCID$_{50}$) 20% animals survived at the end of the 5th week. 3-week survival rate for all the infected groups (low, medium and high inoculum) was 60%. This is significantly longer than most of the current animal model and should be a tremendous advantage for studying pathogenesis and antiviral drugs.

## Materials and methods

### Ethics statement

All animal experiments were approved by the Nanjing University Animal Care Committee and followed the Guide for the Care and Use of Laboratory Animals published by the Chinese National Institutes of Health. The research protocols were conducted in strict accordance and adherence to relevant policies regarding animal handling as mandated under the guidelines from the institutional animal care committee (#2014-SR-079).

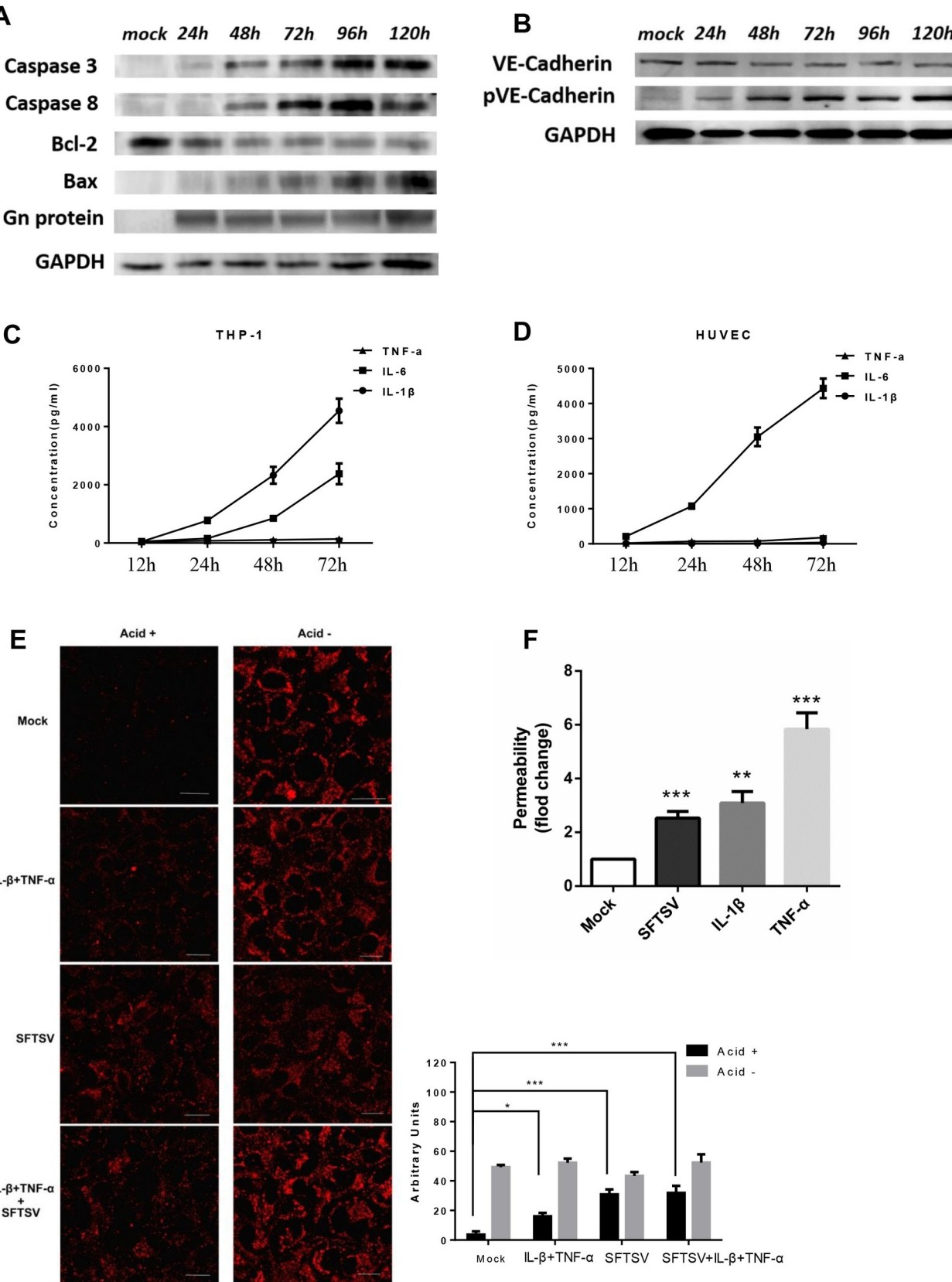

**Fig 6. Apoptosis and VE-cadherin internalization were enhanced in pathogenic SFTSV-infected endothelial cells.** (A) High-dose SFTSV replication in HUVECs inducing apoptosis started from 48 hpi. (B) Phosphorylation of VE-cadherin Tyr685 was induced by SFTSV, and contributes to the proper induction of vascular permeability *in vitro*. (C and D) 3 proinflammatory cytokines released from SFTS virus–infected cells. Supernatant levels of cytokines were quantified by ELISA assays. IL-1β, interleukin 1β; IL-6, interleukin 6; TNF-α, tumor necrosis factor α. (E) Immunofluorescent staining of HUVECs infected with SFTSV or SFTSV-inducible inflammatory cytokines. Following acid washing, the red fluorescent region indicates VE-cadherin endocytosed at higher rates in the virus infected HUVECs. The results were quantified by counting average fluorescence intensity using Image J software. The images are 400×. (F) Confluent HUVECs were infected in triplicate with pathogenic SFTSV at an MOI of 10 or proinflammatory cytokines or mock infected. Either SFTSV alone or SFTSV-inducible inflammatory cytokines could induce endothelial hyperpermeability *in vitro*. Data are shown as mean±SEM of three independent experiments. ($^*$p <0.05, $^{**}$p <0.01, $^{***}$p <0.005).

## SFTSV virion isolation and antibodies against SFTSV

The SFTSV of subtype E-JS-2013-24 was provided by Jiangsu Provincial CDC, Nanjing, China. The virus was passaged in Vero E6 cells [43]. $TCID_{50}$ was determined using methods described previously [1]. Camel and Rabbit antibodies against SFTSV were obtained by immunization with soluble SFTSV Gn protein expressed in mammalian cell line and donated by Y-clone China.

## Humanized NCG mice

Immunodeficient NOD-Prkdc$^{em26Cd52}$Il2rg$^{em26Cd22}$/Nju (NCG) mice were purchased from the GemPharmatech Co., Ltd of Nanjing. HuPBL mice were generated as described previously [44]. Briefly, $2 \times 10^7$ human PBMCs were injected intraperitoneally (i.p.) into each of 4- to 6-week-old NCG mice. PBMC engraftments were evaluated at the 21$^{st}$ day after the transplantation by FACS.

## SFTSV infection of HuPBL-NCG mice

Animal study was approved by the Institutional Animal Welfare Committee and conducted in compliance with biosafety guidelines. HuPBL mice (*n* = 5) were injected intraperitoneally with various amount of SFTSV, ranging from $10^3$ to $10^5$ $TCID_{50}$ (50% tissue culture infective dose) and five mock-infected mice were used in parallel as controls. At each time point of days 1, 3, 7, 14, 21, and 28, the viral load in blood and the blood routine indexes were analyzed after exsanguinating blood through the orbital sinus. Different tissues were collected immediately after animal sacrifice.

## Viral load determination by quantitative real-time PCR

The viral RNA in mouse sera and tissues were extracted using Quick-RNA Viral Kit (ZYMO Research, CA, USA) or TRIzol reagent (Thermo Fisher Scientific, MA, USA) and cDNAs were generated by reverse transcription using RT-PCR Prime Script Kit (Takara, CA, USA) or HiScript III RT SuperMix for qPCR (+gDNA wiper) Kit (Vazyme, Nanjing, China). Viral copy numbers were determined by real-time RT–PCR with an L segment-based SFTSV-specific primer set (SFTS-LF': TGGTGGATGTCATAGAAGGC and SFTSV-LR': GAGTAATCCTCTTGCCTGCT). Viral copy numbers were calculated as a ratio with respect to the standard control [14]. Gene expression of relative fold change, recorded as cycle threshold (Ct), was normalized against an internal control (GAPDH). Real-time PCR was performed using a SYBR Green Supermix (Bio-Rad, CA, USA) and determined on Applied Biosystem 7500 (Life Tech, NY, USA) according to the manufacturer's procedure [1,45].

## Immunofluorescence and confocal microscopy

Samples of SFTSV-infected HuPBL mouse tissues were fixed in 4% (wt/vol) paraformaldehyde (PFA) and embedded in paraffin according to standard procedures and the embedded tissues

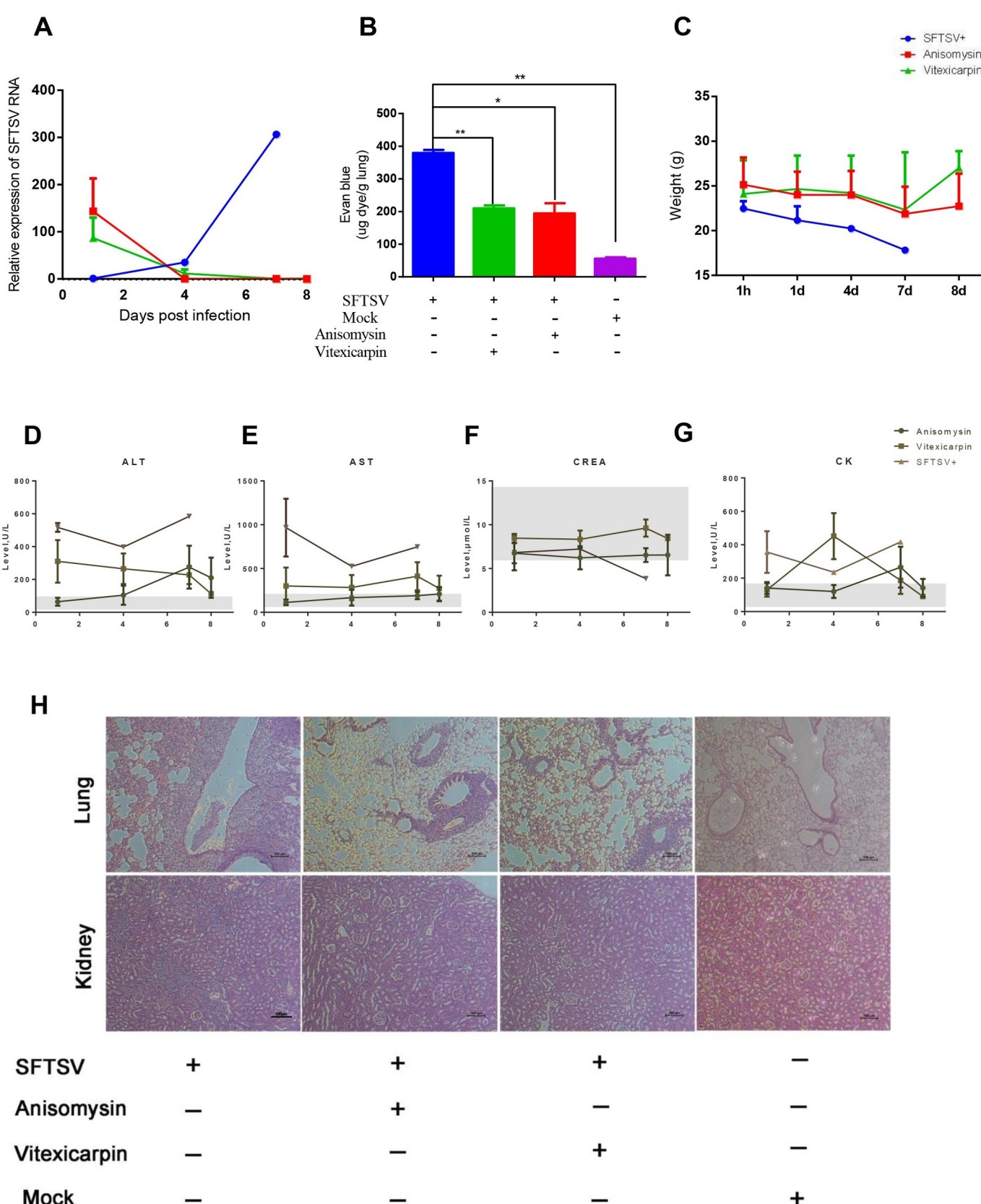

**Fig 7. Antiviral drugs stabilize the vasculature and reduce mortality after SFTSV infection *in vivo*.** (A) Antiviral drugs anisomysin and vitexicarpin inhibited the viral replication. (B) Anisomysin and vitexicarpin reduced vascular permeability in the lungs. (C) Anisomysin and vitexicarpin stabilized the significant change in the body weight of mice *in vivo* (D-G) Kinetics of hematological and biochemical parameters in virus infected mice following treated with antiviral drugs were performed at the indicated time points until animals were euthanized. In each graph, gray shadows indicate the physiological reference range of the parameter according to the virus uninfected mice. (H) Immunohistochemistry staining showed that antiviral drugs improved the pathological condition in lung and kidney tissues of SFTSV infected mice. The Scale bars in H&E images are 20 μm. Abbreviations: ALT, alanine aminotransferase; AST, aspartate aminotransferase; CREA, creatinine; CK, creatine kinase. Data are shown as mean±SEM of three independent experiments. (*$p < 0.05$, **$p < 0.01$).

were sectioned and dried for 3 days at room temperature. Thin sections (4 μm) of tissues were deparaffinized with xylene and ethanol. Immunofluorescence staining was performed to detect viral antigen as previously described [46,47]. SFTSV-specific camel polyclonal Abs, prepared in our laboratories, were used as the primary antibody for detecting SFTSV antigens. After blocking with 3% BSA (wt/vol) in assay buffer, the sections were stained with primary and secondary antibodies. Gn-specific camel polyclonal Abs (Y-Clone, Suzhou, China) and FITC-conjugated anti-camel IgG Fc fragment (Life Tech, NY, USA) were used to probe SFTSV. After virus staining, rabbit anti-F4/80 mAb (eBioscience, CA, USA) and Alexa Fluor 647-conjugated anti-rabbit IgG (Life, NY, USA) were used to probe macrophages. Finally, cell nucleus was stained with 4',6-diamidino-2-phenylindole (DAPI) and the triple-fluorescence stained tissue sections were sealed by fluorescence antiquencher reagent (Invitrogen, CA, USA) and observe under an Olympus confocal microscope FV3000.

## Tissue histology

5 weeks after SFTSV infection, mice were sacrificed by diethyl ether asphyxiation. Lungs, spleens, livers and kidney were inflated with 4% (wt/vol) PFA and the PFA-fixed tissues were processed, embedded in paraffin, sectioned at 4 μm, and stained with hematoxylin and eosin (H&E). The severity of histologic pathology was quantified according to the methods described [48]. The spleen pathological injury was scored according to the severity of reduced cellularity in the red pulp, which could be divided into 5 grades: 1, fully depleted cellularity; 2, lymphocyte islands were absent but scattered lymphocytes visualized; 3, substantially decreased number and size of lymphocyte islands; 4, slightly decreased number and size of lymphocyte islands; 5, normal cellularity with many lymphocyte islands [16]. Besides, megakaryocyte infiltration in the spleen reflects the virus infection rate. The average number of megakaryocytes were counted from 10 randomly chosen view fields for each sample. In lung tissues, the slides were also scored for pathological severity of disease on a scale of 0 to 5 grades according to the alveoli integrity and the inflammatory factors invasion. Scored categories are for macrophages, lymphocytes, neutrophils, pleuritis, bronchiolar inflammation, perivascular inflammation, eosinophils, edema, and epithelial necrosis. The global inflammation score is according to interstitial inflammation, bronchiolar inflammation, and perivascular inflammatory response. Scores were averaged, and rounded to the nearest whole number. With that they were compartmentalized for low/none (0 or 1), medium (2 or 3), or high (4 or 5) [49]. The liver pathology was quantified as the percentage of incidences that showed hepatocyte ballooning degradation or necrosis in liver sections. The kidney pathology was quantified as the percentage of the total glomeruli within a field that showed glomerular hypercellularity, mesangial thickening, or congestion, counted from 10 randomly chosen 20× view fields for each sample.

## Flow cytometric analysis

PBMCs were isolated from mouse peripheral blood using BD FACS Lysing Solution (BD Biosciences, CA, USA) according to the manufacturer's protocol. Cell pellets were re-suspended in 2 ml of 1X lysing solution and incubated at room temperature for 15 minutes. After lysis,

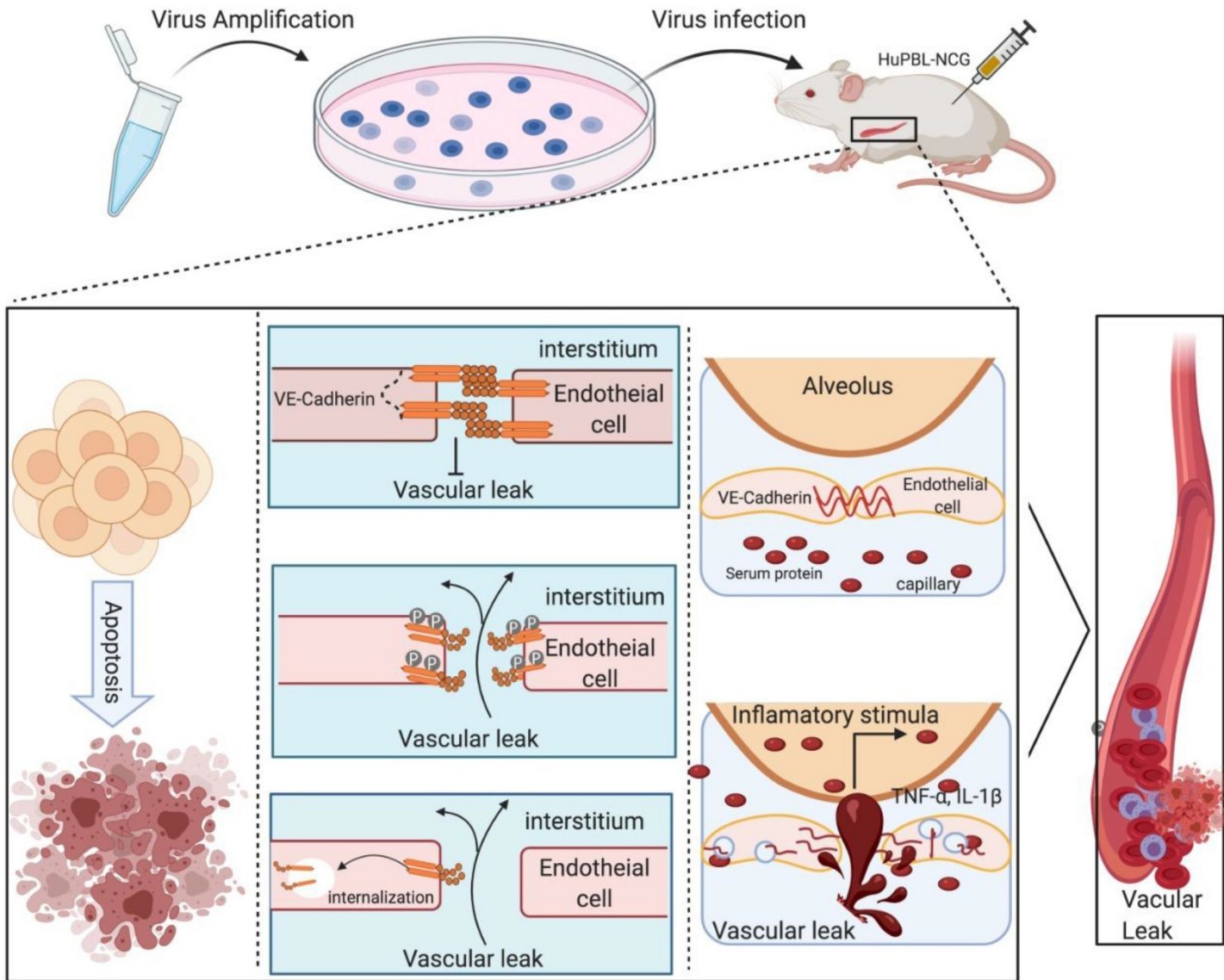

**Fig 8. Graphical illustration showing the experimental pipeline and mechanism of haemorrhage.** The machanism of hemorrhage that cells apoptosis, membrane protein endocytosis and cytokine stimulation are likely involved in inducing endothelial hyperpermeability.

samples were pelleted at 700 g in a microcentrifuge for 5 min at room temperature, and stained with 100 μl monoclonal antibody mixture containing 2 μl of each anti-human CD45-APC and CD14-Percyp-cy5.5 (Biolegend, CA, USA). The monocyte labeling was performed for 60 min at 4˚C. Before intracellular staining, cell pellets were washed with 1 ml phosphate buffered saline (PBS) supplemented with 2% fetal bovine serum and permeabilized using BD Cytofix/Cytoperm Fixation/Permeabilization Kit. The fixed/permeabilized cells were thoroughly resuspended in 50 μl of BD Perm/Wash buffer containing a pre-determined optimal concentration of a Gn-specific camel antibody and subsequently stained with FITC-conjugated anti-camel IgG Fc. Pelleted cells were resuspended in 200 μl of wash buffer and analyzed on flow cytometer (NovoCyte Flow Cytometer, ACEA). Cells were firstly gated for human CD45 before analyzing other parameters.

## ELISA

Secreted cytokines in cell culture supernatants and sera from infected mice were quantified by ELISA kits (Excell Bio, Shanghai, China) per manufacturer's protocols. Supernatants from stimulated HUVECs and THP-1 cells or SFTSV infected mouse sera were collected at indicated time points and subjected to immunoassay with commercial ELISA kits for mouse or human IL-1β, IL-6, IL-4 and TNF-α.

## Immunoblotting analysis

Whole-cell lysates were prepared in RIPA lysis buffer and measured for protein concentration by BCA protein assay kit (Life Technologies, USA) to equalize protein loading. Proteins were resolved in SDS-PAGE gels and transferred to a PVDF (polyvinylidene difluoride, Millipore, USA) membrane at 100 V for 60 min (Bio-Rad). All membranes were blocked in 3% BSA in TBS with 0.5% Tween 20 (TBST; pH 8.0; Sigma) and probed with indicated antibodies in antibody dilution reagent solution (Life Technolies, MD, USA) at 4˚C overnight, and then incubated with either IRDye Fluor 800-labeled IgG (1:10000 926–32210) or IRDye Fluor 680-labeled IgG (1:10000 926–68071) secondary antibody (Li-COR Bioscience). The images were scanned and quantified by densitometric analysis on Li-COR Odyssey Infrared Imager. Primary antibodies includes antibodies specific for caspase 3, caspase 8, Bcl-2, BAX, and GAPDH (Proteintech), antibodies specific for phospho-VE-cadherin (Tyr685; Cell Signaling) and antibodies specific for VE-cadherin (CST). Rabbit anti-SFTSV-Gn antibodies were kindly provided by Y-Clone, Ltd.

## Internalization assay

VE-cadherin internalization was performed as previous described with the follow modification [50]. In brief, HUVECs were seeded onto chamber slides and cultured for 72 hours. Two days post infection, HUVECs were starved overnight in 0.5% BSA without growth factors, and subsequently treated with cytokines combination (10 ng/ml IL-1β and 10 ng/ml TNF-α), or mock treated (control). Endothelial cells were followed by incubation with anti-VE-cadherin antibody (CST) for 1 h at 4˚C. Redundant antibody was removed with ice-cold PBS by washing coverslips, and the cells were shifted to a 37˚C $CO_2$ incubator for 1 h to allow VE-cadherin internalization. Cells were then left untreated or followed by washing with a mild acid solution (2 mM PBS-glycine, pH 2.0; three times for 5 min) to remove unbound antibody. Next, cells were washed with PBS, fixed in 4% paraformaldehyde for 10 min, permeabilized with 0.25% Triton X-100 for 5 min, and incubated with goat anti-rabbit Alexa 594–conjugated secondary antibody (Life, NY, USA) for 1 h [51,52]. Coverslips were mounted using fluorescence antiquecher reagent (Invitrogen) and examined using an Olympus confocal microscope FV3000 with ImageJ software measured. Results were representative of 3 to 5 independent experiments

## Evans blue permeability *in vivo* and *in vitro*

*In vivo* **vascular permeability assays.**   Vascular permeability assays in the lung was assessed with EBA as described [53]. 2 weeks after $10^5$ $TCID_{50}$ SFTSV infection, mice were treated with an intravenous injection of EBA (20 mg/kg). EBA was allowed to circulate for 1 hour. Mice were deeply anesthetized and perfused with saline plus 5 mM EDTA. Lungs were subsequently excised, weighed, and homogenized in 1.5 ml of PBS. Formamide (3 ml; Invitrogen) was added, and the samples were incubated at 60˚C overnight to extract Evans blue dye. Those samples were then treated by centrifuge, and supernatants were analyzed by

spectrophotometry at both 620 and 740 nm (Infinite M200). The absorbance was normalized as described [53] and transformed to microgram Evans blue dye per gram wet weight of lungs. Data are presented as SEM of at least four mice per condition.

**In vitro endothelial permeability assays.** To measure permeability changes of endothelial cells in vitro, HUVECs were seeded onto vitronectin (10 g/ml)-coated Costar Transwell inserts (12-mm diameter and 3-μm pore size; Corning) at a cell density of 2 x 10$^4$ cells and grown in DEMED with 10% of fetal bovine serum for 2 days to form mature monolayers. Confluent endothelial monolayers were infected in quadruplicate either with SFTSV (MOI = 10), treated with cytokines IL-1β (10 ng/ml) and TNF-α (10 ng/ml), or left untreated (mock). After 48 h postinfection, culture media were replaced and washed with colorless PBS. Then, 500μl Evans blue dye (0.5mg/ml) was added to the upper chamber of monolayers and incubated for 5 min. Liquid (100 μl) from the lower chamber were transferred to a 96-well plate and were read using a spectrophotometric microplate reader at both 620 and 740 nm.

## Inhibition of SFTSV-induced HUVEC permeability

HUVECs were grown in Transwell plates and infected as described above. SFTSV (MOI = 10) were co-cultured with different natural phytochemicals for 60min at 37˚C prior to infecting cells. Evans Blue dye were added to the upper chamber of cells in the presence or absence of Vitexicarpin (0.025 μM to 2.5 μM) or Anisomysin (0.001 μM to 0.1 μM) at 3 days post infection. The permeability of monolayers was evaluated by quantitating the appearance of Evans Blue in the lower chamber as described above. Both of these two natural phytochemicals were solubilized in DMSO and stored at -80˚C. An identical amount of DMSO diluent was added to DMEM (10% fetal bovine serum) as negative control experiments. The results are derived from two to five independent experiments and reported as the mean ± the standard deviation of the fold permeability for each group. The statistical significance was analyzed by two-tailed Student's *t*-test and expressed as a P value. $P < 0.05$ was considered to be statistically significant.

## Endothelial cell migration

**Scratch wound migration assay.** Scratch wound migration tests were carried out using 1 ml pipette tip to scratch on the surface of full confluence cells in the middle of a 12-well plate cultured with or without SFTSV infection. The wound-healing process was recorded at 24h, 48h and 72h after the scratch under a 40 x objective. The area of wound recovered versus that of the original wound was identified as the wound healing rate.

**Tube formation assay.** The tube formation assay was used to assess angiogenesis *in vitro*. In brief, 100 μl Matrigel (Corning) was planted into precooled 24-well plate on ice and incubated for 30 min at 37˚C. Then, the HUVECs were digested and 1 x 10$^5$ cells were transferred to the wells in 500 μl medium. Cells were treated with either SFTSV infection, VEGF (100 ng/ml) addition, or left untreated (negative). After incubation at 37˚C for 24 h and 48 h, tube formation was observed and captured with Olympus inverted microscope. Tube length were quantified with ImageJ software (National Institutes of Health).

## Statistical analysis

All quantitative data are presented as means ± SEM. Significance between specific data sets is described in the respective figure legend and was performed using ANOVA or independent-samples *t*- test in GraphPad Prism 5 software.

## Supporting information

**S1 Fig. SFTSV infection in different immunocompromised mice.** (A) HuPBL-SCID mice flow cytometry analysis of human CD45+, CD3+ and CD8+ cells. (B) Effect of on SFTSV infected AG6 mice. The number of viral RNA copies in blood of SFTSV-infected mice, survival curve and weight change curve. (C) IL-2 knockout in a SCID background NCG mice remain refractory to SFTSV infection, while be susceptible to virus infection after humanized with PBMCs. (D) Body temperature during SFTSV infection of HuPBL-NCG mice. The results presented that no obvious difference was observed among different virus titers infection groups. Data are shown as mean±SEM of three independent experiments.(****p <0.001).
(TIF)

**S2 Fig. Megakaryocytes staining and SFTSV track *in vivo*.** (A) Immunofluorescence assay detected the number of megakaryocytes increased significantly in the spleen, Bar = 100 μm. (B) Fluorescence labled Gn specific antibody detected SFTSV distribution *in vivo*. After intravenous injection of fluorescence labled Gn specific antibody in HuPBL mice, the Far-infrared fluorescence of whole animal or various organs was acquired at the indicated time points by NightOwl LB 983. Data are shown as mean±SEM of three independent experiments.(**p <0.01).
(TIF)

**S3 Fig. Flow cytometry analysis of three cells contact manners.** Combining CD45-APC stained THP-1 with Gn-FITC stained SFTSV-infected cells, which demonstrated all THP-1 were infected with SFTSV and THP-1 significantly improve the virus infection of RAW264.7 cells *in vitro* via direct cell-cell contact. Data are shown as mean±SEM of three independent experiments. (****p <0.001).
(TIF)

**S4 Fig. Confocal images of Rhesus Macaques Spleen.** Detection and colocalization of SFTSV antigens in tissues of Rhesus Macaques Spleen. The confocal image shows viral Gn protein (in green) accumulating in anti–platelet cell adhesion molecule 1 (PECAM-1; CD31) vascular endothelium cells (in red), Scar bar, 50 μm.
(TIF)

**S5 Fig. SFTSV induce apoptosis of endothelial cell proinflammatory cytokines inhibitors improve endothelial cell permeability.** (A) Annexin V-FITC/PI Apoptosis Detection Kit test the apoptosis of virus-infected HUVEC, SFTSV could directly induce apoptosis of endothelial cell. (B) The measurement using TUNEL/Caspase 3 immunofluorescence staining showed that endothelial apoptosis could be observed in virus infected mouse model, Scar bar = 50 μm. (C) Low-dose (MOI = 1) virus infection without cells apoptosis. No significant elevated apoptosis level were observed within 5 days of infection. (D) Dynamic change of three proinflammatory cytokine inhibitors increased the amount of VE-cadherin protein persist on the cell membrane, Bar = 20 μm. (E) Three proinflammatory cytokines inhibitors suppressed SFTSV-directed permeabilizing responses at physiologic concentrations. Data are shown as mean ±SEM of three independent experiments. (*p <0.05, **p < 0.01).
(TIF)

**S6 Fig. Antiviral drugs test *in vitro*.** (A-C) The molecular structure of anisomysin and vitexicarpin and the effect of these two drugs on SFTSV, the images are 200x. (D-E) Cytotoxicity of anisomysin and vitexicarpin to HUVEC and Vero cells. HUCEV and Vero cells were incubated with serial concentrations of two drugs and cell viability was measured by CCK-8 kit after 48 h. (F-G) Permeability changes of endothelial cells on HUVECs after anisomysin and

vitexicarpin treated. Data are shown as mean±SEM of three independent experiments. (**p < 0.01, ***p <0.005, ****p <0.001).
(TIF)

**S7 Fig. CBA multi-human cytokines test.** Anisomysin and Vitexicarpin decreased the plasma level of multi-human inflammatory factor. Data are shown as mean±SEM of three independent experiments. *p < 0.05.
(TIF)

## Author Contributions

**Conceptualization:** Shijie Xu, Zhiwei Wu.

**Data curation:** Shijie Xu, Xilin Wu.

**Formal analysis:** Shijie Xu, Xilin Wu, Zhiwei Wu.

**Funding acquisition:** Xilin Wu, Zhiwei Wu.

**Investigation:** Shijie Xu, Na Jiang, Xilin Wu, Zhiwei Wu.

**Methodology:** Shijie Xu, Na Jiang, Waqas Nawaz, Bingxin Liu, Fang Zhang, Ye Liu, Xilin Wu.

**Project administration:** Zhiwei Wu.

**Resources:** Xilin Wu, Zhiwei Wu.

**Software:** Shijie Xu.

**Supervision:** Zhiwei Wu.

**Visualization:** Shijie Xu.

**Writing – original draft:** Shijie Xu.

**Writing – review & editing:** Shijie Xu, Xilin Wu, Zhiwei Wu.

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
