## [Decision Letter · Decision Letter 0]

3 Feb 2021

Dear Dr. wu,

Thank you very much for submitting your manuscript "Infection of humanized mice with a novel phlebovirus presented pathogenic features of sever fever with thrombocytopenia syndrome" for consideration at PLOS Pathogens. As with all papers reviewed by the journal, your manuscript was reviewed by members of the editorial board and by several independent reviewers. In light of the reviews (below this email), we would like to invite the resubmission of a significantly-revised version that takes into account the reviewers' comments.

Please also address the following points:

1.) It is important to clearly indicate how NGC mice were reconstituted with PBMC and how inter-donor variation is/can be accounted for in your model. Please also indicate this caveat in your text.

2.) Beyond permeability, direct measurement of endothelial apoptosis in your mouse model is warranted (ie TUNEL/active caspase strain of infected tissues).

3.) The text indicates that infection had a suppressive effect on lung endothelial permeability with 3.2X less leakage, but your data indicates the opposite? Please address. Please provide images with increased magnification for Figure 5. Tissue images with Evans blue and/or albumin/IgG to visbaly demonstrate leakage would strength conclusions.

4.) Minor: please add scale bars to all images, individual data points to all graphs and number of replicates in all legends. “Flod” change should be corrected to Fold change.

We cannot make any decision about publication until we have seen the revised manuscript and your response to the reviewers' comments. Your revised manuscript is also likely to be sent to reviewers for further evaluation.

Sincerely,

Kellie Jurado

Guest Editor

PLOS Pathogens

Sara Cherry

Section Editor

PLOS Pathogens

Kasturi Haldar

Editor-in-Chief

PLOS Pathogens

orcid.org/0000-0001-5065-158X

Michael Malim

Editor-in-Chief

PLOS Pathogens

orcid.org/0000-0002-7699-2064

It is important to clearly indicate how NGC mice were reconstituted with PBMC and how inter-donor variation is/can be accounted for in your model. Please also indicate this caveat in your text.

Beyond permeability, direct measurement of endothelial apoptosis in your mouse model is warranted (ie TUNEL/active caspase strain of infected tissues).

The text indicates that infection had a suppressive effect on lung endothelial permeability with 3.2X less leakage, but your data indicates the opposite? Please address. Please provide images with increased magnification for Figure 5. Tissue images with Evans blue and/or albumin/IgG to visbaly demonstrate leakage would strength conclusions.

Minor: please add scale bars to all images, individual data points to all graphs and number of replicates in all legends. “Flod” change should be corrected to Fold change.

Reviewer's Responses to Questions

**Part I - Summary**

Reviewer #1: In this manuscript , authors innovatively developed a SFTSV infection model based on the Hu-PBL-NCG mice that recapitulates many pathological characteristics of SFTSV infection in humans. Authors further revealed that SFTSV infection increased the vascular permeability of endothelial cells by promoting tyrosine phosphorylation and internalization of the adhesion molecule vascular endothelial (VE)–cadherin. In addition, authors found that both virus infection and pathogeninduced exuberant cytokine release dramatically contributed to the vascular endothelial injury. However, the following comments and questions need to be addressed.

Reviewer #2: The authors reconstituted SCID NCG mice with human PBMC (hPBMC-NCG mice) and infected the mice with severe fever with thrombocytopenia syndrome virus (SFTSV). The authors determined whether the infected mice recapitulated SFTS and could be used as a model system to evaluate antiviral drugs against SFTSV.

Rationale for using hPBMC-NCG mice as SFTSV infection model has not been explained well in this manuscript.

**Part II – Major Issues: Key Experiments Required for Acceptance**

Reviewer #1: (No Response)

Reviewer #2: 1. The authors cited studies reporting immune incompetent mice supporting SFTSV. NCG is immune incompetent mouse.

1) What did the authors hypothesize to observe by infecting NCG mice with SFTSV, based on the characteristics of NCG mice?

2) Why did the authors expect to see enhancement of SFTSV infection in hPBMC-NCG mice whose immune systems had been reconstituted with human immune cells, but not in NCG mice?

2. There is no mention of how human PBMC was secured for the study.

3. The infection model depends on the role of human PBMCs in NCG mice. Although the model looks feasible as a proof of concept, the system might work variably according to the human donor of PBMC, since severity of SFTS appears variable judging from seroprevalence studies. Without consistency guaranteed, could it be called a model system?

4. The reconstituted mice worked for the authors to observe recapitulation of SFTS. But, would it work next time? It is not clear if each hPBMC-NCG mouse was reconstituted with PBMCs from each different person or all hPBMC-NCG mice used were reconstituted with PBMCs from a same person. If the latter was the case, reproducibility of the system might need to be tested using PBMCs from another person.

5. Why did the authors not use murine endothelial cells instead of HUVEC for mechanistic characterization of the observed leakage (by Evans Blue) and influences of cytokines of Fig 5 and Fig 6?

**Part III – Minor Issues: Editorial and Data Presentation Modifications**

Reviewer #1: 1. Authors developed SFTSV infection HuPBL-NCG mice model, it is necessary to detect body temperature and blood pressure during SFTSV infection.

2. Considering the influence of transplanted human PBMC, it is better to have a negative control (uninfected) group during SFTSV infection in Fig. 1FG. And what about Platelet counts and WBC counts from low dose virus infected HuPBL-NCG mice during infection? Can this low dose virus infected mice eliminate virus and detect blood cell recovery?

3. Authors found decreased cellularity and obvious lymphocyte depletion in the red pulp (RP) and the number of megakaryocytes increased significantly in the spleen by immunohistochemistry. It would be helpful if authors could detected by immunofluorescence or flow cytometry at the same time.

4. Fig. 3A flow chart needs to add the information of horizontal and vertical axes. Fluorescence and corresponding epitope should be marked on the Fig. 3E.

5. Transwell culture system can effectively prevent direct cell contact. So it would be fine if authors change Fig. 3CD cytokine+SFTSV group by Transwell THP1+SFTSV group. Authors found THP-1 significantly improve the virus infection ability among mouse RAW264.7 cells in vitro. It is advisable that to verify it in vivo, such as adding NCG mouse infection group to Fig. 3E.

6. In this manuscript , could SFTSV directly induce apoptosis of endothelial cell? Do they have similar effects on other endothelial cells?

Reviewer #2: Fig 1: switching positions of C and D is recommended.

Fig 3D: About RAW + THP-1, it cannot be said that THP-1 enhanced SFTSV infection in RAW, since there was no control for THP-1 alone; the data should be reinterpreted.

Fig 5B: It seems that description in lines 238 – 244 appears opposite to Fig 5B

Line 29; While previous models -> Previous models

Line 56; with respect of cells apoptosis -> with respect to apoptosis

Line 146; may -> may be

Line 177; widely -> wide

PLOS authors have the option to publish the peer review history of their article (what does this mean?). If published, this will include your full peer review and any attached files.

Reviewer #1: No

Reviewer #2: No
---

## [Editor Report · Decision Letter 1]

26 Apr 2021

Dear Prof. Wu,

We are pleased to inform you that your manuscript 'Infection of humanized mice with a novel phlebovirus presented pathogenic features of sever fever with thrombocytopenia syndrome' has been provisionally accepted for publication in PLOS Pathogens.

Best regards,

Kellie Jurado

Guest Editor

PLOS Pathogens

Sara Cherry

Section Editor

PLOS Pathogens

Kasturi Haldar

Editor-in-Chief

PLOS Pathogens

orcid.org/0000-0001-5065-158X

Michael Malim

Editor-in-Chief

PLOS Pathogens

orcid.org/0000-0002-7699-2064
---

## [Editor Report · Acceptance letter]

6 May 2021

Dear Prof. Wu,

We are delighted to inform you that your manuscript, "Infection of humanized mice with a novel phlebovirus presented pathogenic features of sever fever with thrombocytopenia syndrome," has been formally accepted for publication in PLOS Pathogens.

Best regards,

Kasturi Haldar

Editor-in-Chief

PLOS Pathogens

orcid.org/0000-0001-5065-158X

Michael Malim

Editor-in-Chief

PLOS Pathogens

orcid.org/0000-0002-7699-2064